# Influence of Operating Temperature on the Service Life of Aluminum Extrusion Dies

**DOI:** 10.3390/ma15196656

**Published:** 2022-09-26

**Authors:** Rafał Hubicki, Maria Richert, Piotr Łebkowski

**Affiliations:** 1Grupa Kęty S.A., 32-650 Kęty, Poland; 2Management Department, AGH University of Science and Technology, 30-067 Kraków, Poland

**Keywords:** aluminum extrusion, matrix to aluminium extrusion, nitriding, surface heat treatment, structure

## Abstract

The article investigates the effect of temperature and annealing time on the surface quality of WNLV nitrided steel used for the production of dies for extrusion of aluminum alloys. Eight annealing variants were tested, differing in the total annealing time at temperatures of 460 °C and 590 °C. The results show the effect of the annealing time on the width of the diffusion layer, which increased with the increasing length of the annealing time. The hardness decreased as the annealing time increased. It was found that annealing of the steel causes its oxidation. The oxide layer formed consisted of two layers, more specifically, an Fe_2_O_3_ oxide layer and a lower Fe_3_O_4_ oxide layer adhering to the steel surface. The surface of sections pressed on oxidized matrices was tested. The roughness of the surface of the oxide layers was also tested. The research revealed that an increase in the surface roughness of the oxides deposited on the matrices causes an increase in the roughness of the extruded sections. These results can potentially be used to improve the efficiency of the extrusion process and the quality of the extruded sections.

## 1. Introduction

The global aluminum extrusion market size was valued at USD 57.29 billion in 2020 and is expected to grow at a compound annual growth rate (CAGR) of 7.8% from 2021 to 2028.

It is predicted from 2021 to 2028 to reach USD 104.15 billion by 2028.

Low costs and weight of extrusion products contribute to their rising demand in various industrial applications, mainly in the automotive, aerospace and defense sectors.

The growth of the aluminum market is related to the growth of the market of tools and equipment used for extrusion of aluminum profiles. In particular, the dies that shape the extruded profiles are an important element of any extrusion process. Increasing the service life of the extrusion dies will translate into significant financial savings of plants producing aluminum profiles. Work on extending the service life of the dies is of great economic importance. In particular, there is a great potential for improvement in their heat treatment and operating conditions. Nitriding of matrices after quenching and tempering treatment is a commonly used surface hardening operation, followed by nitriding [1].

Before extrusion, the dies are heated to a temperature of over 400 °C, and then, in the extrusion process, they work at a temperature of about 560 °C. The materials from which the dies are made are tool steels subjected to thermal improvement and nitriding. In the extrusion process, the dies undergo structural changes which result in changes in properties. The most serious is the reduction of the hardness of the die orifice. This occurs as a result of the diffusion of elements under the influence of the temperature at which the matrices work, which results in structural changes. The processes of dissolving carbides and nitrides take place. Another phenomenon related to the annealing and operation of the matrix at high temperatures is the oxidation of the matrix surface. Usually, the annealing of the dies before extrusion is the result of the practice used in the enterprise and takes place in standard furnaces. As a result of prolonged keeping of the dies in the soaking furnaces, uncontrolled changes in the structure appear, adversely affecting their functional properties. It is unfavorable to lower the hardness of the die orifice below 1000 HV units, which makes it necessary to regenerate them via secondary nitriding, the purpose of which is to renew the properties and re-harden the surface. The effects of multiple nitriding were investigated in the study [2]. It was found that multiple-nitriding treatment on H13 steel has a significant effect on compound layer thickness and its phases, diffusion zone depth and its microstructure, hardness–depth profile, and nitride case depth. It was found that excessive cumulative nitriding time during multiple-nitriding treatment results in greater nitride depth and a significant increase in hardness with deeper effect due to the dense and deeper precipitation of nitrides in the diffusion zone. Multiple-nitrided samples show oxidation and porosity in the near-surface part of the nitrided layer due to the interaction of iron with oxygen of the air upon decomposition of iron nitrides in the compound layer during re-nitriding. This results in reduced toughness and hardness in the near-surface part of the nitrided layers. The influence of repeated nitriding on the surface structure of matrices and new methods of nitriding were also investigated in the works of Borowski et al. [3,4]. The effect of overheating of the matrix and the extrusion ingot on the formation of streak defects was investigated by Zhu et al. [5], who considered influencing factors involved in various processing steps such as billet quality, extrusion process, die design and etching process. Kugler et al. [6] pointed out that the quality of the matrix-bearing surface as well as its tribology and geometric conditions determine the quality of the product surface. The matrix is repeatedly heated and re-nitrided and is subject to significant structural changes. After the originally performed thermal improvement treatment, subsequent annealing and operation at high temperatures change the properties; in particular, it leads to the coagulation of carbides and grain growth, which results in a general reduction in the strengthening of the matrix material. The research on the defects of gas nitrided matrices was carried out in the work of Birol [7].

Akhtar et al. [2] found that the thickness of the single-nitrided layer is 80 μm, for the double nitrided layer is 100 μm, and that for the triple-nitrided layer is 115 μm. The result suggests an increase of the nitrided layer width with the prolongation of exposure at high temperature during nitriding. They also found that the hardness and nitrided depth were increased with increasing the nitriding time during the multiple-nitriding treatment.

In work by Duma et al. [8], a diffusion-hardened surface was examined with different characterization techniques. The effect of nitriding time on diffusion layer thickness, tensile properties, impact energy, and wear resistance properties was investigated. It has been shown that increasing the nitriding time enables thicker diffusion layers that modify the adhesive’s wear mechanism to abrasive. Increased nitrided layer thickness resisted friction much more compared to less nitrided specimens. Fernández et al. [9] investigated the influence of die material impact on extrusion parameters and the quality of extruded sections. They found the extrusion process is affected by different parameters such as core diameter, billet height, friction, extrusion ratio, ram speed, temperature and die semi-angle. The simulation studies covered the influence of parameters on die wear. It has been shown that the matrix material has a significant impact on the extrusion process. Muro et al. [10] showed the influence of the matrix material in the hot stamping process on the process of its wear. Terčelj and Kugler [11] pointed out that optimizing geometry and technological parameters allows the life of matrices to be extended. A broad discussion of the failure of aluminum extrusion dies has been performed in work by Arif et al. [12]. Research on the impact of speed frames on the surface quality of the extruded aluminum alloy 6063 was carried out in the study by Park et al. [13]. Surface defects occurred only at ram speed of 12 mm∙s^−1^. The results suggest the dependence of the appearance of surface defects in the product on the extrusion speed.

An aluminum profile is formed via the die orifice (die hole). In addition, the quality of products and extrusion productivity often depend on die performance [14]. Normally, an aluminium. extrusion die is heated to the nearest billet temperature at about 425 °C–450 °C. Furthermore, the significant parameter is the die preheat time. To prevent underheating or overheating of the die, the die preheat time also needs to be strictly controlled.

The control of the annealing time is also important due to the die oxidation process, which adversely affects the smoothness of its surface. Publication [15] draws attention to the unfavorable oxidation effect of the matrices during multiple annealing. A total of 16 different types of iron oxide were distinguished. Generally, however, on the surface of iron and its alloys, Fe_2_O_3_ and Fe_3_O_4_ oxides are the most common [16].

The surface quality of the die orifice significantly affects the surface phenomena during extrusion. A study of the impact of the smoothness of the matrix on the extrusion process was conducted in paper [17]. Using different surface topographies and lubricants on the die cone, the frictional conditions for correct extrusion were investigated. A polished die cone, commonly used in extrusion, is used as a reference. The best product was extruded taking into account the analyzed quality factors, profile integrity and profile surface quality.

The effect of extrusion parameters (extrusion speed and temperature), die geometry and coating material on the operation life of extrusion dies were investigated in work [18]. Greater roughness of products extruded on dies not coated with TiB_2_ layers was found. Authors of work [19] concluded that many defects in the extruded products occur because of the conditions of the dies and tooling. The mechanisms of die surface wear were investigated in the work of Zhao et al. [20].

This article focuses on the study of the matrix wear areas, in particular the die orifice. The changes occurring as a result of heating at the temperatures corresponding to the heating of the extrusion dies and at the temperatures corresponding to the extrusion process were investigated and analyzed. Experimental simulation of these processes made it possible to trace the structural changes and changes in properties resulting from the annealing of nitrided WNLV tool steel (EU, EN-55NiCrMoV7, USA-L6). The influence of the quality of the die surface on the quality of the extruded sections from 6063 aluminum alloy was also analyzed.

## 2. Materials and Methods

The test specimens were made of WNLV steel (alloy tool steel for hot work—Standard: ISO 4957) ( EN-55NiCrMoV7 Shanghai Unite Steel Trading Co., Ltd., Shanghai, China), which is used for die inserts for presses and forging machines, anvils for hammers and presses with large dimensions and deep cuts and hammer dies of medium dimension sizes. Table 1 shows the chemical composition of WNL steel used to make dies and samples.

The samples were thermally tempered in accordance with the procedure used for this type of steel: hardening at a temperature of 840 °C–890 °C and tempering at a temperature of 500 °C–600 °C. Then, they were subjected to nitriding in a Nitrex furnace at a temperature of 460 °C for 6 h.

After nitriding, a physical simulation of the operation processes was carried out, consistent with the processes of exploitation of dies, intended for extrusion of aluminum alloy 6063 profiles used in the company.

The dies were used to extrude 6063 aluminum alloy sections with the composition given in Table 2. The ram speed of extrusion of 6063 alloy was 9 mm/s.

Table 3 contains the applied variants of annealing of the samples with the specification of the temperature and the annealing time. Table 4 shows the annealing times used at particular temperatures for individual annealing variants. Eight different annealing variants were carried out with the total annealing time of the samples from 6 to 28 h.

In all variants, annealing was carried out at a temperature of 460 °C for 4 h, which simulated the process of annealing the matrix before entering into the press. Annealing at a temperature of 590 °C simulated the extrusion process itself, which, depending on the test variant, lasted from 2 to 6 h. A temperature of 590 °C was adopted in the tests based on production data for the current year.

The samples were tested for hardness and structure. The samples were cut perpendicular to the nitrided surface using a Struers cutter with 250 mm diameter Discotom-10 66A25 discs. They were then sanded on sandpaper and polished according to the Struers technique for steel, followed by etching in Nital reagent.

The structure studies were carried out using an OLYMPUS inverted metallographic microscope, type GX53 (OLYMPUS, Warsaw, Poland). The tests were carried out on sections perpendicular to the nitrided surface. The samples were cut with a Struers Discotom-10 cutter. The specimens were prepared according to the Struers technique (MD-Piano 220 (water). B) MD-Allegro (DP-Susp. P 9 µm). C) MD-Largo (DP-Susp. P 3 µm). D) MD-Dac (DP-Susp. P 3 µm). E) MD-Chem (OP-S NonDry). Polished surfaces were etched with Nital 3% reagent.

In addition to these studies, the structure was observed using scanning and transmission electron microscopy. The samples were observed in scanning electron microscopes (SEM) at an accelerating voltage of 5 kV in the SE and BSE observation modes. In the BSE mode, both mass contrast and indicative contrast were obtained. The observations were carried out on Hitachi SU8000 and Hitachi SU70 microscopes, (Hitachi, Tokyo, Japan). During the research, a microanalysis of the chemical composition was also performed using a ThermoFisher (Thermo Fisher Scientific Inc. (NYSE: TMO), Pittsburgh, PA, USA) energy dispersion spectrometer. The microanalysis of the chemical composition allowed for the identification of elements in the oxide coating and carbide precipitation. Moreover, the structure was investigated using the EBSD method—electron back scatter diffraction.

Structure observations were also made using a Hitachi HD2700 Transmission Scanning Microscope with an accelerating voltage of 200 kV. The sample for STEM observation was cut out via the lift-out technique using a Hitachi NB5000 double-beam ion electron microscope. The STEM observations were performed in diffraction and mass contrast. For SEM observation, the samples were prepared as cross-sections through the nitrided layer. They were cut with a slowly rotating diamond saw. Then, using an ion cutter (Hitachi IM4000), about 40–70 micrometers of material was sprayed with a wide beam of argon ions on a surface of about 0.5 × 0.5 mm^2^. In this way, it was possible to obtain a cross-sectional area through the nitrided layer and the substrate material. Sputtering made it possible to obtain mass and indicative contrasts and prevented stresses during material removal. The only artifact that arose during the sputtering was a delicate relief on the surface, created by material discontinuities and surface irregularities.

The microhardness of the samples was tested on polished specimens cut perpendicular to the nitrided surface using a Shimadzu semi-automatic microhardness tester, type HMV-G21. X-ray examinations were carried out using a Bruker D8 ADVANCE X-ray diffractometer with filtered radiation Cu Kα (λ = 0.154056 nm) at room temperature. Test parameters were as follows: voltage −40 kV, current −40 mA, angle 2Θ from 15° to 120°. The XRD patterns were analyzed using Bruker EVA software and a PDF-2 database (from the International Centre for Diffraction Data). X-ray examinations were carried out selectively at small angles to the surface (greasing), which made it possible to identify the chemical composition of the oxides layer.

Research on the influence of annealing time on the surface quality of extruded aluminum profiles was carried out. The tests were carried out in laboratories and on the technological line in the press shop of Grupa Kęty S.A. The furnace also contained samples of WNLV steel, from which the dies were made. EN-AW 6063 aluminum alloy profiles were extruded on the annealed dies, from which samples were taken for roughness testing. Measurements of the deposited oxide thickness and roughness were performed on the steel samples. The roughness was assessed by measuring the microscopic unevenness of the surface profile on the cross section perpendicular to the surface of the samples. Rz was determined on an inverted metallographic microscope at a magnification of 500X; the length of the measuring section was 200 µm, and the length of the sampling section was 40 µm.

Measurements of the temperature distribution on the die cross-section during annealing in a furnace simulating the annealing conditions of the die before the extrusion process were carried out in order to optimize the annealing temperature. The matrix was annealed in a Castool furnace at a temperature of 460 °C. The 12 holes (6 on each side of the die) were made in the die, at a distance of 20 mm (on the die diameter). The temperature was measured with a K type thermocouple made by CZAH, TKP-1-4500-3-2-1-1-2-1.

## 3. Results

The diffusion layer width for the applied annealing variants, which was determined on metallographic specimens using a light microscope, is shown in Figure 1 as a function of the total annealing time of the samples including the total annealing time at 460 °C and 590 °C.

The tests showed that the samples v1 and v3 had the smallest diffusion layer width, and they were annealed at 460 °C and 590 °C, respectively, for 12 and 6 h. Samples v1 and v3 were not kept at a temperature of 20 °C, which differs from the other samples. Additionally, they also had some of the shortest annealing times at 460 °C and 590 °C (Table 4). Presumably, these factors limited the diffusion of nitrogen in the steel. Figure 2 shows the structure of the diffusion layer of sample v2, and Figure 3 shows the sample structure of v3. 

There was an oxide layer on the surface of the samples. The thickness of the oxide layer in the tested samples is shown in Figure 4. It can be seen that the thickness of the oxide layer increased with increasing the total annealing time of the samples. The annealing tests of WNLV steel, as well as the oxidation tests of AISI 304 stainless steel [21], revealed the presence of a layered structure of the oxide with a different chemical composition. Scanning microscopy studies revealed that the oxide layer has a complex structure. There were two layers in the oxide coating. An outer zone and an inner zone adjacent to the surface of the samples were distinguished. Such a two-zone structure of the oxide layer was observed in all tested samples. An exemplary image of the oxide layer is shown in Figure 5 for the v2 sample after 22 h of annealing. The two zones that make up the oxide layer are clearly visible. In the literature, a complex structure of oxide layers is often reported, the thickness of which increases with the annealing time [22,23,24]. The molecules of oxygen are absorbed from the atmosphere, react with the iron and nucleated oxides. The oxide nuclei grow from the surface of the steel and form a layer with time.

Chemical composition tests, carried out using the X-ray method, revealed two types of iron oxides Fe_2_O_3_ (hematite) and Fe_3_O_4_ (magnetite) in the oxide layer. X-ray diffraction of the near-surface area showed that in the outer layer (near the sample surface) only Fe_2_O_3_ oxide (hematite) was present. The deeper layer, contained the second Fe_3_O_4_ oxide identified in the research, i.e., magnetite (Figure 6). A clearly visible delamination of the oxide layer was observed, separating the upper oxide zone from the lower one adjacent to the steel substrate (Figure 7). The outer oxide layer was found to be more compact, with a clearly visible crystal structure made of Fe_2_O_3_ magnetite. The lower zone was made of hematite Fe_3_O_4_ and contained an additional, lighter silicon phase, which is shown in Figure 7. The presence of silicon particles was identified via the linear examination of the chemical composition using the EDX method on a scanning microscope (Figure 8). The lower oxide layer growing into the grain structure of the steel was characteristic. Penetration of the oxide into the steel proceeded along the grain boundaries of the substrate (Figure 9). The two-layer structure of the oxide on the surface of WNLV steel probably results from a multi-stage annealing at two temperatures of 460 °C and 590 °C. The kinetics of the growth of the oxide changes with increasing temperature, as well as its morphology. This causes changes in the chemical composition, manifested by the diversity of structure of growing oxide layers.

The microhardness tests showed that only samples v1 and v3 had a hardness higher than 1000 HV units (Figure 10). The value of 1000 units is the limiting minimum hardness value for tools suitable for extrusion of aluminum alloys. Samples v1 and v3 simultaneously had the narrowest diffusion layer (Figure 1). The total annealing times for the samples v1 and v3 are respectively 12 and 6 h.

There is therefore a correlation between the total annealing time, the hardness and the width of the diffusion layer. The higher the hardness, the narrower the diffusion layer and the shorter the total residence time of the samples at high temperatures. The obtained data constitute an important indication for the correct preparation of dies for aluminum extrusion.

It was found that the quality of the die surface influences the surface quality of the extruded sections. In particular, the absence of an oxide on the surface of the sizing strip is of great importance in achieving a smooth surface of the extruded section. Studies have shown that extending the die annealing time increases the thickness of the oxide formed on the die surface and increases its roughness, which affects the surface quality of aluminum profiles. Figure 11 shows the changes in the roughness of the sections and the roughness of the oxide as a function of the annealing time. The correlation between the oxide and its roughness and the roughness of extruded aluminum sections was investigated. With the increase of the annealing time, not only the thickness of the oxide increased (Figure 12), but also its roughness. After 4 h of annealing, the roughness of the oxide was R_z_ = 1.0, and after 32 h of annealing, R_z_ = 2.8. Accordingly, the roughness of aluminum profiles increased from the value of R_z_ = 0.9 for the annealing time of 4 h to R_z_ = 1.5 for the annealing time of 32 h. The obtained result clearly shows that extending the die annealing time increases the roughness of the oxide deposited on the die surface, which affects the surface quality of the extruded sections, increasing its roughness.

Figure 12 Shows the changes in the roughness of the oxide depending on its thickness. As the annealing time increases, the oxide thickness increases from 1.3 µm for 4 h annealing time to 5.7 µm after 32 h of annealing. At the same time, its roughness increases.

The presented data constitute a microscopic assessment of the surface structural changes of the matrix and indicate the significant importance of its oxidation and smoothness on the surface quality of the sections.

The extrusion dies were annealed until reaching 460 °C. The temperature was measured on the cross-section with thermocouples placed on the radius of the die (Figure 13). On the basis of the tests performed, it was found that the dies themselves reached the set temperature of 460 °C within 2 h of annealing (Figure 14). After this annealing time, no oxide was found on the die surface. The same experiment carried out on dies in a housing showed that the dies reached a temperature of 460 °C after 4 h of annealing (Figure 15). It was the shortest annealing time that was used in the research on the surface quality of the dies. This indicates that there is sufficient time to obtain the desired temperature of the extrusion dies. At the same time, this time corresponds to the most favorable parameters of the tested samples.

## 4. Discussion

The conducted research was aimed at the physical simulation of the actual operating conditions of the dies in the process of extrusion of aluminum alloys, the closest to the actual technological process. Standard nitrided samples in a Nitrex furnace had a starting hardness of 1169 HV. It was found that as a result of annealing, after various variants of the combination of temperatures of 460 °C and 590 °C, the hardness of most of the samples decreased below 1000 units. Only samples v1 and v3 after heat treatment showed a hardness above 1000 units, respectively sample v1 hardness was 1023 HV (with standard deviation 26.99) and sample v3 hardness was 1025 HV (with standard deviation 11.22). Analyzing the obtained results, the hardness measurements were compared to the total annealing time of the samples, and it was found that the decrease in hardness correlates with the increase in the total annealing time at temperatures of 460 °C and 590 °C. The samples with hardness above 1000 HV had the shortest annealing times. Respectively, the total annealing time of these samples was 12 h for the v1 sample and 6 h for the v3 sample. Taking into account the standard deviation, only the v3 sample with the lowest annealing time met the hardness criterion.

The obtained result indicates the necessity to select the annealing regime in such a way that the dies are heated to the desired temperature in the shortest possible time.

As a result of long-term annealing, the samples were covered with an oxide coating. The structure of the oxide coating was complex. It consisted of an outer layer of magnetite Fe_2_O_3_ and a layer of hematite Fe_3_O_4_ adhering to the substrate. Discontinuities were found between the layers. This suggests that the upper oxide layer can be easily crushed during the extrusion of the sections.

It was found that the presence of the oxide on the die surface is of significant importance as it affects the surface quality of the extruded sections.

The conducted research shows that with the increase of the annealing time, the thickness of the oxide and its roughness increase at the same time. It was found that the increase in the roughness of the oxide on the surface of the die influences the increase of the roughness of the extruded sections.

The established correlations revealed the importance of the impact of the surface quality of the dies on the quality of the extruded sections. This draws attention to the problems of tools and their correct use.

In particular, compliance with the recommended heat treatment parameters, annealing times and care for ensuring an appropriate protective atmosphere that can be used under given conditions during annealing of dies is crucial in achieving the best production results.

Borowski et al. [25] found that the increase in the extrusion speed of aluminum sections leads to a reduction in their roughness. In the context of these results, the research carried out in this article suggests that the higher extrusion speed increases the probability of chipping rough oxides on the die surface, which contributes to a smoother surface of the extruded sections. The high depth of the matrix without oxides also increases the smoothness of the aluminum film, which forms on the surface of the matrix calibration strip to reflect the unevenness of its surface [26]. On the other hand, research on the influence of the extrusion speed on the roughness of aluminum sections, carried out by Romański and Burdek [27], showed a reduction in roughness in the speed range of 10–30 m/min. At a speed of 40 m/min, the roughness of the sections increased. The correlation of the results of work conducted on the issue of the quality of extruded products leads to the conclusion that certain process and preparation parameters of the tools may prevent some of the unfavorable phenomena lowering the quality of the extruded products.

Interest in the issues of dies is significant due to the costs associated with their production and heat treatment, as well as in terms of the impact of the surface quality of the matrix on the quality of the surface of the extruded products. The surface quality of the dies depends on the operating conditions (temperature, operating time, extrusion speed). The present work introduced the surface phenomena occurring on the surface of the die calibration strip, which influences the surface quality of the extruded aluminum sections.

## 5. Conclusions

Based on the research, the following conclusions were drawn:It was found that with the increase of the annealing time, the width of the diffusion layer increases from about 170 µm (6 h of annealing) to about 200 µm (about 20 h of annealing).The tests have shown that after 4 h of annealing, an oxide layer is formed on the surface of the dies. It was found that the oxide layer has a two-zone structure and consists of two types of oxides. In the outer zone (near the sample surface) there was only Fe_2_O_3_ oxide (hematite). The deeper layer, contained the second Fe_3_O_4_ oxide identified in the research—magnetite.Research has shown that the increase in the surface roughness of the oxide deposited on the die correlates with the increase in surface roughness of the extruded sections.

## Figures and Tables

**Figure 1 materials-15-06656-f001:**
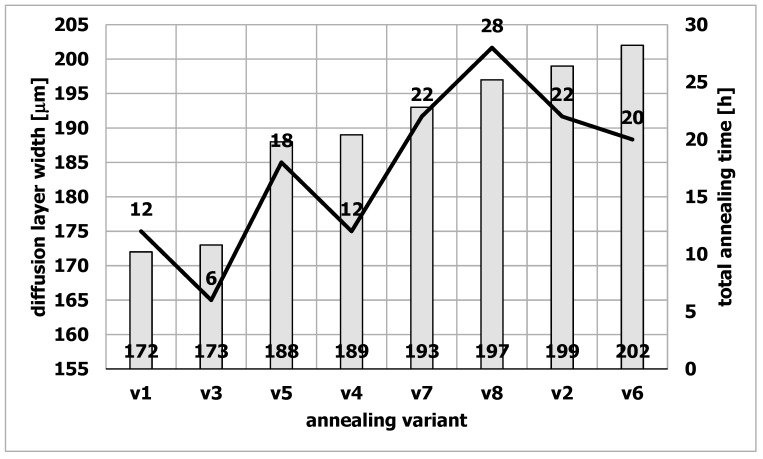
The width of the diffusion layer against the background of the total annealing time of the samples at the temperatures of 460 °C + 590 °C.

**Figure 2 materials-15-06656-f002:**
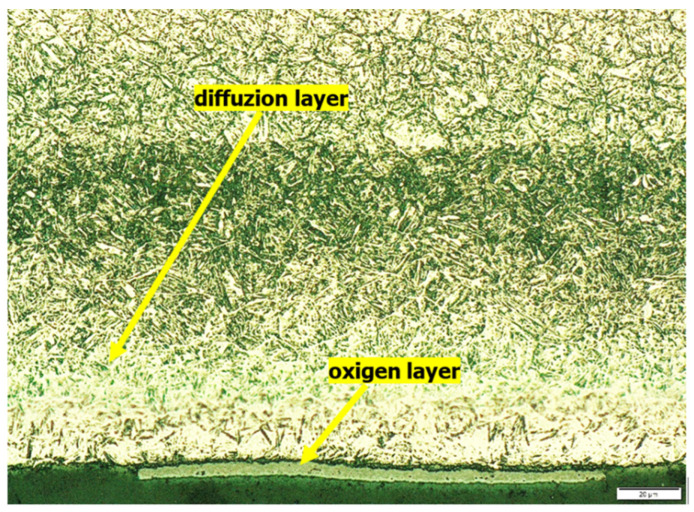
The structure of the v2 sample cross-section, magnification 2500×.

**Figure 3 materials-15-06656-f003:**
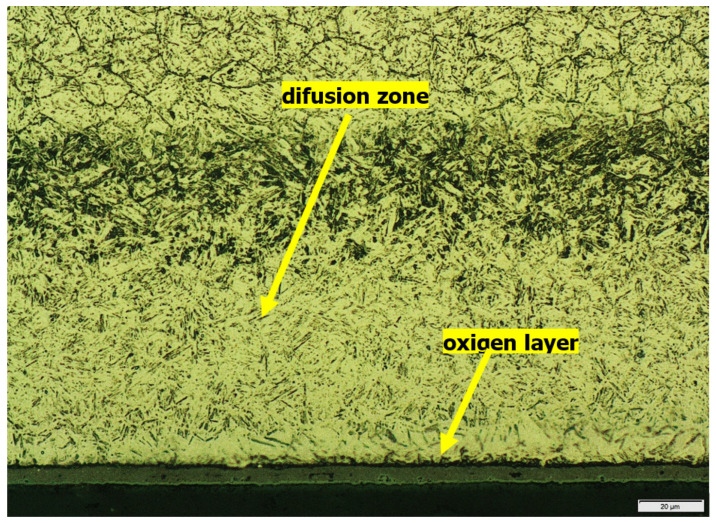
The structure of the v3 sample cross-section, magnification 2500×.

**Figure 4 materials-15-06656-f004:**
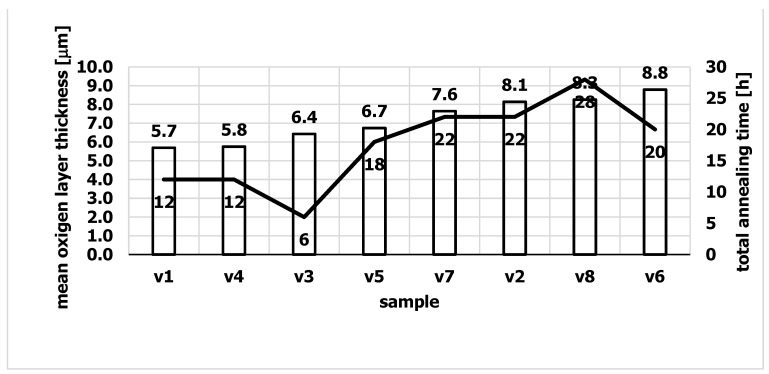
Thickness of the oxide layer of the annealed samples.

**Figure 5 materials-15-06656-f005:**
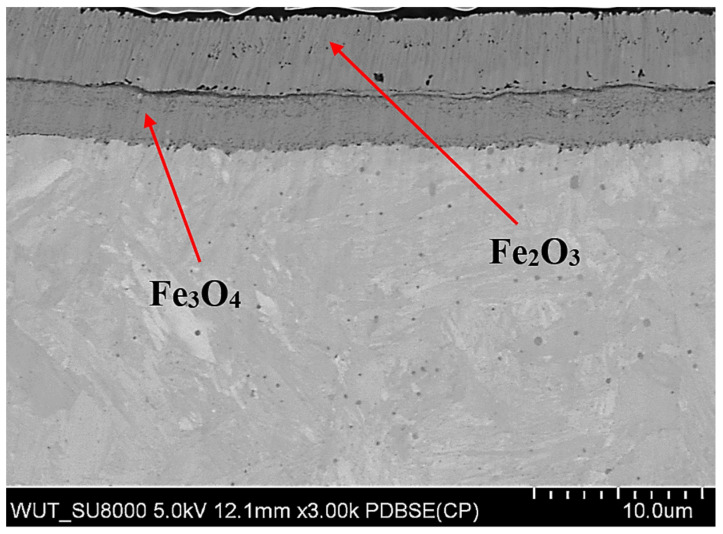
Structure of the v2 sample with the oxide layer (total annealing time 22 h, SEM) magnification 3000×.

**Figure 6 materials-15-06656-f006:**
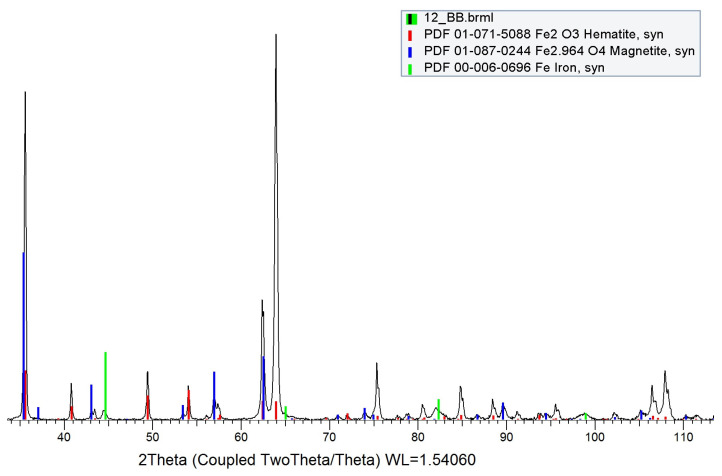
Result of Bragg-Brentano RTG analysis of oxygen layer showing kind of oxigens, sample v9.

**Figure 7 materials-15-06656-f007:**
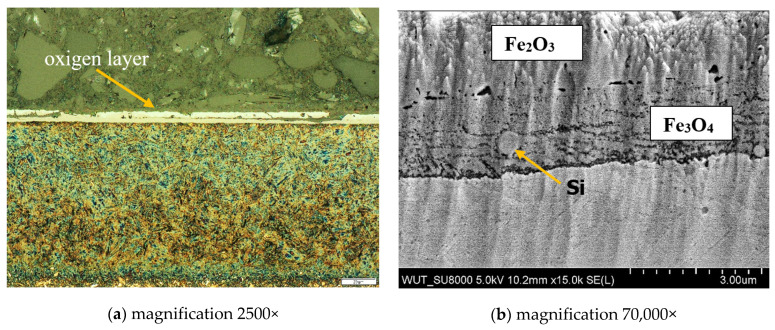
Two-zone oxide layer on the surface of the v1 sample (12 h annealing).(**a**) microstructure of steel with oxide layer indicated by arrow, magnification 2500×; (**b**) enlarged fragment of the oxide layer, the upper zone of the layer contains Fe_2_O_3_, the lower zone contains Fe_3_O_4_, silicon precipitation was identified in the lower zone of the oxide layer.

**Figure 8 materials-15-06656-f008:**
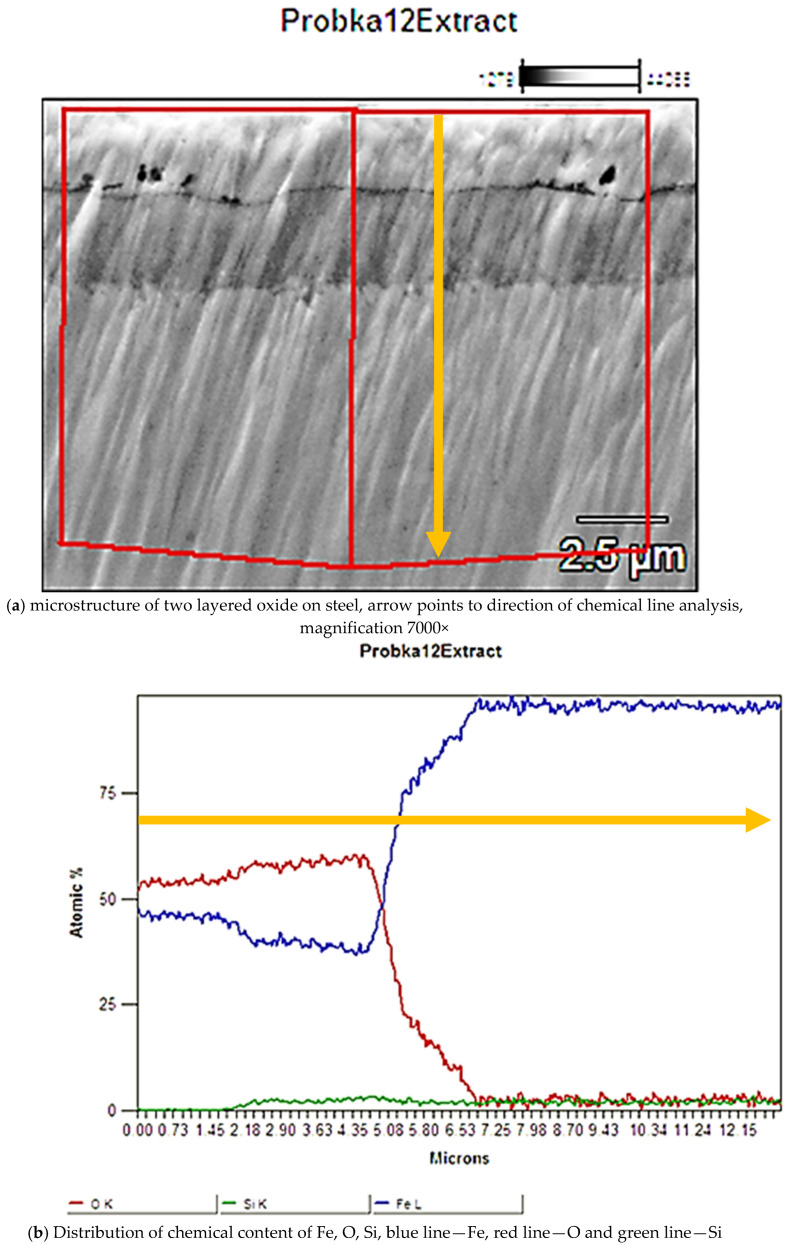
Linear analysis of the chemical composition of the oxide layer, sample v7 (22 h annealing),.(**a**) microstructure of two layered oxide on steel, the red lines and arrow points to the scope of the chemical composition test, magnification 7000×; (**b**) distribution of content of Fe, O and Si in the oxide layer, blue line—Fe, red line—O and green line—Si; the direction of arrow from left to right corresponds to the direction from the surface into the specimen; oxygen disappears after about 7 μm from surface, the green Si line in the area where oxygen is present indicates its presence in the oxide layer.

**Figure 9 materials-15-06656-f009:**
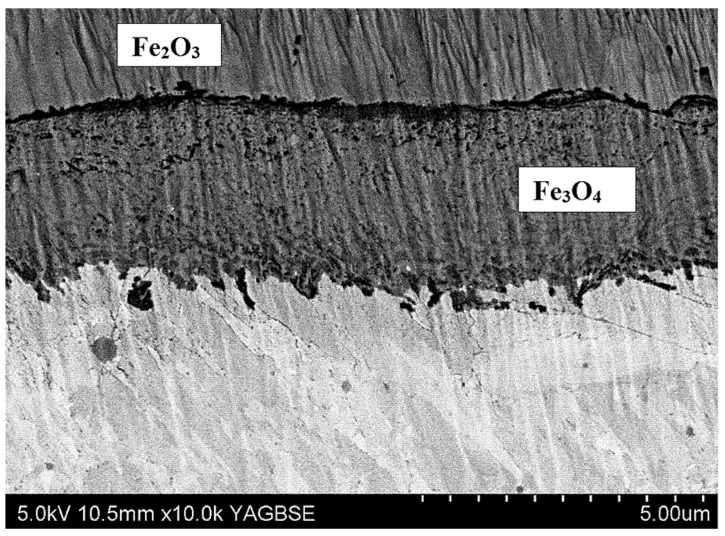
Oxide layer growing into the substrate, v7, two sections with Fe_2_O_3_ and Fe_3_O_4_ content are identified (total annealing time 22 h) magnification 10,000× (SEM).

**Figure 10 materials-15-06656-f010:**
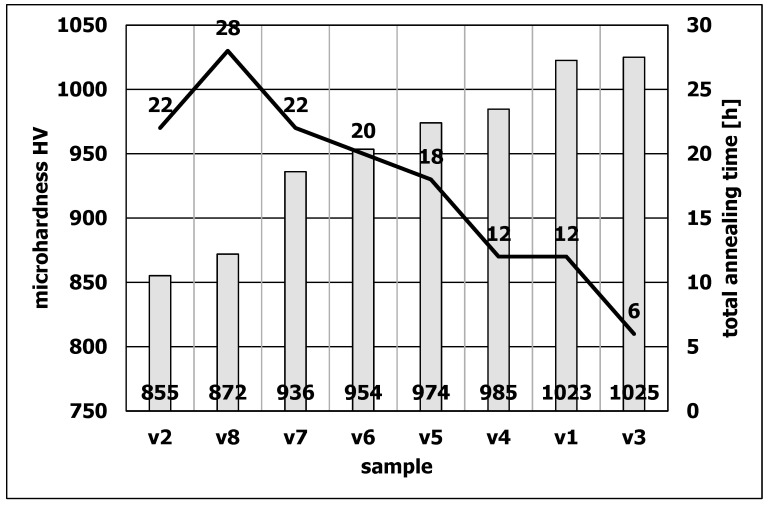
Microhardness of samples with a marked annealing time line for individual heat treatment variants.

**Figure 11 materials-15-06656-f011:**
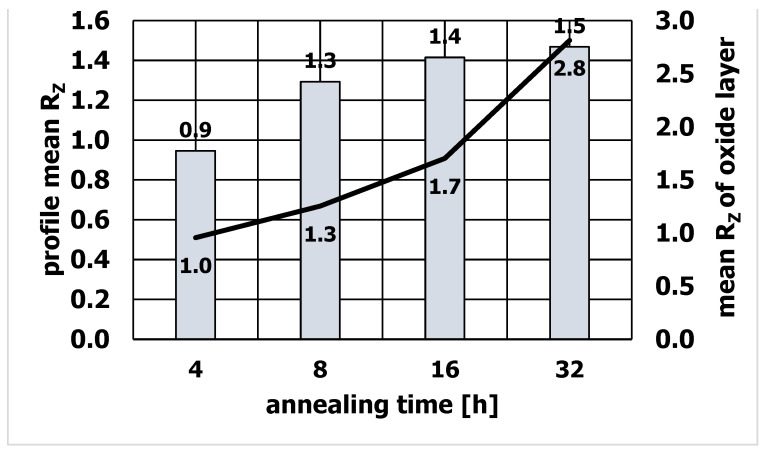
Roughness of extruded aluminum profiles and the oxide deposited on the die.

**Figure 12 materials-15-06656-f012:**
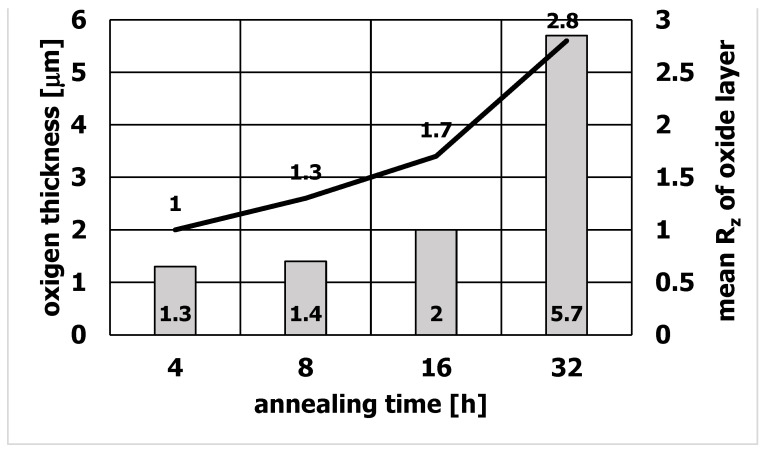
Thickness and roughness of the oxide deposited on the WNLV steel matrix.

**Figure 13 materials-15-06656-f013:**
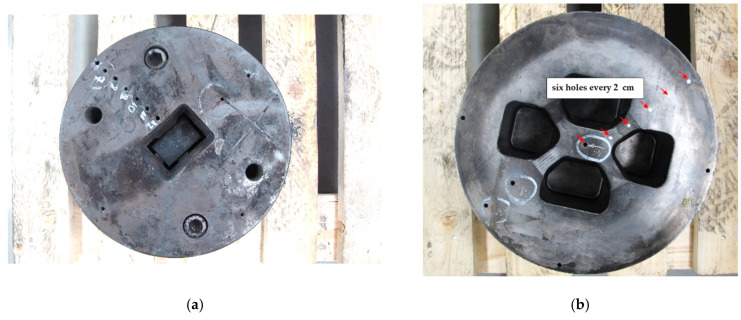
Dies with marked places of thermocouples that measure the achieved temperature during annealing at 460 °C. (**a**) first example of die; (**b**) second example of die.

**Figure 14 materials-15-06656-f014:**
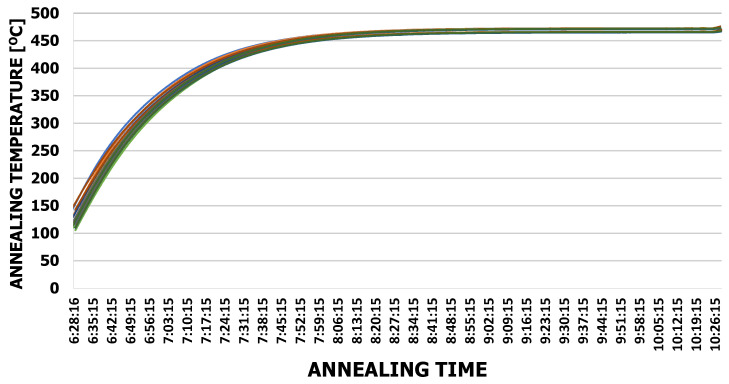
Annealed die temperature graph without housing.

**Figure 15 materials-15-06656-f015:**
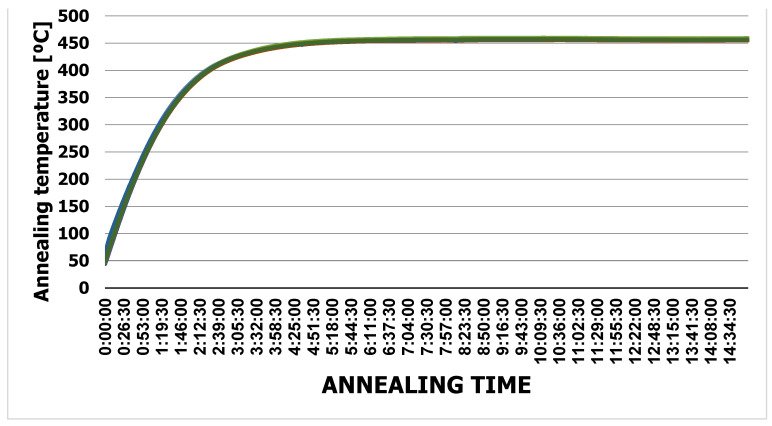
Temperature graph of the annealed die in the housing.

**Table 1 materials-15-06656-t001:** Chemical composition of used WNLV steel.

C	Si	Mn	P	S	Cr	Mo	V
0.39	0.99	0.31	0.018	0.003	5.07	1.22	0.93

**Table 2 materials-15-06656-t002:** Chemical composition of the test melt from 6063 aluminum alloys.

Alloy Grade	Si	Fe	Cu	Mn	Mg	Zn	Ti	V
6063	0.53	0.21	0.02	0.04	0.48	0.01	0.01	0.01

**Table 3 materials-15-06656-t003:** Variants of annealing of nitrided samples made of WNLV steel for electron-microscopic examinations.

The Heat Treatment	Heat Treatment Cycle
Variant 1/12 h	460 °C/4 h–590 °C/2 h–460 °C/4 h–590 °C/2 h
Variant 2/22 h	460 °C/4 h–590 °C/2 h–460 °C/4 h–590 °C/2 h–460 °C/4 h–590 °C/6 h
Variant 3/6 h	460 °C/4 h–590 °C/2 h
Variant 4/12 h	460 °C/4 h–590 °C/2 h–20 °C/12 h–460 °C/4 h–590 °C/2 h
Variant 5/18 h	460 °C/4 h–590 °C/2 h–20 °C/12 h–460 °C/4 h–590°C/2 h–20 °C/12 h–460 °C/4 h–590 °C/2 h
Variant 6/20 h	460 °C/4 h–590 °C/6 h–20 °C/12 h–460 °C/4 h–590 °C/6 h
Variant 7/22 h	460 °C/4 h–590 °C/6 h-20 °C/12 h - 460 °C/4 h–590 °C/2 h–20 °C/12 h–460 °C/4 h–590 °C/2 h
Variant 8/28 h	460 °C/4 h–590 °C/6 h–20 °C/12 h–460 °C/4 h–590 °C/2 h–20 °C/12 h–460 °C/4 h–590 °C/2 h–20 °C/12 h–460 °C/4 h–590 °C/2 h

**Table 4 materials-15-06656-t004:** Annealing times for samples at particular temperatures.

Heat Treatment	Total Annealing Time (h)	Annealing Time at 460 °C (h)	Annealing Time at 590 °C (h)	Holding Time at 20 °C (h)
Variant 1	12	8	4	
Variant 2	22	12	10	
Variant 3	6	4	2	
Variant 4	12	8	4	12
Variant 5	18	12	6	24
Variant 6	20	8	12	12
Variant 7	22	12	10	24
Variant 8	28	16	12	36

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
