# Peer review of "Influence of Operating Temperature on the Service Life of Aluminum Extrusion Dies"

_materials, 2022, doi:10.3390/ma15196656_

Round 1
Reviewer 1 Report
File attached

Author Response
Response to the Reviewer's 1 comments
All comments included in the review have been included in the revised article and marked in red
The chemistry of the extruded aluminum alloy has been added to the Materials and Methods section.
The dies were used to extrude 6063 aluminum alloy sections with the composition given in Table 2. The ram speed of extrusion 6063 alloy process was 9mm/s.
Table 2. Chemical composition of the test melt from 6063 aluminum alloys
|
Alloy grade |
Si |
Fe |
Cu |
Mn |
Mg |
Zn |
Ti |
V |
|
6063 |
0,53 |
0,21 |
0,02 |
0,04 |
0,48 |
0,01 |
0,01 |
0,01 |
The tests showed that the samples v1 and v3 had the smallest diffusion layer width, and they were annealed at 4600C and 5900C, respectively, for 12 and 6 hours. Samples v1 and v3 were not kept at the temperature of 200C, which differs from the other samples. Additionally, they also had some of the shortest annealing times in 4600C and 5900C (Table 4.). Presumably, these factors limited the diffusion of nitrogen in the steel.
The two zones that make up the oxide layer are clearly visible. In literature it is often reported complex structure of oxide layers, which thickness increases with the annealing time [23,24,25]. The molecules of oxygen are absorbed from the atmosphere, reacted with the iron and nucleated of oxides formation of a thin layer which grows to a thicker layer with the increasing time of annealing.
The two-layer structure of the oxide on the surface of WNLV steel probably results from a multi-stage annealing at two temperatures of 4600C and 5900C. The kinetics of the growth of the oxide changes with increasing temperature, as well as its morphology. This causes changes in the chemical composition manifested by the diversity of structure growing oxide layers.
Drawings have been corrected in article text. An explanation has been added in Figure 8.
The description of Rz in Fig. 12 has been changed.
A speed framework is given in the Materials and Methods chapter. It was 9mm/s.
Deleted "research was carried out on the optimization of the annealing time" and inserted another description.

Reviewer 2 Report
This research manuscript studies the influence of annealing matrices on their structure and properties. Although the topic is very interesting and the basis of the investigation is adequate, some deficiencies are observed that should be corrected to consider publication:
1) The format of template has been modified. Please check it thoroughly.
2) Abstract not “Abstracts”
3) The grade Celsius simbol is ⁰C
4) The number of section is always “1”
5) The references number are very vague. More over, in the Introduction section. A revision of the Works performed by other authors is necessary. This topic have received many attention lately, for example you can find some recent papers dealing with selection of die material, heat treatment and the impact on the process:
· Muro, M.; Aseguinolaza, I.; Artola, G. Die Material Selection Criteria for Aluminum Hot Stamping. J. Manuf. Mater. Process. 2021, 5 (1), 15. https://doi.org/10.3390/jmmp5010015
· Fernández, D.; Rodríguez-Prieto, A.; Camacho, A.M. Selection of Die Material and Its Impact on the Multi-Material Extrusion of Bimetallic AZ31B–Ti6Al4V Components for Aeronautical Applications. Materials 2021, 14 (24), 7568. https://doi.org/10.3390/ma14247568
· Qamar, S.Z.; Pervez, T.; Chekotu, J.C. Die Defects and Die Corrections in Metal Extrusion. Metals 2018, 8 (6) , 380. https://doi.org/10.3390/met8060380
6) Table 2 is not in the same format of the rest of tables. I would recommend to change this table by a T-t figure with the different variants. It would help to see more visually the treatment feautures
7) Legends in Figure 3, 5 and 9 are cut.
8) The format of references is not adequate. Use the MDPI style. Review thoroughly. Links to SS NN (like RG) are not accepted. Only the citation of the article published in a reputed journal. In addition it is neccesary to complete the references
Author Response
Response to the Reviewer's 2 comments
All comments included in the review have been included in the revised article and marked in red
Abstract, Celsius symbol, article chapter numbering has been corrected.
The suggested articles are cited in the Introduction. In addition, new additional articles related to the research were cited and highlighted in red in the literature list. Section Introduction has been reworked.
Table 2 has been reformatted.
Figures No 3,5,9 have been improved.
Literature was presented in the MDPI style and revised

Reviewer 3 Report
This work focuses on evolution of microstructure of die treated at different conditions. They are well characterized and the results are interesting. However, there are few points to be clarified as below:
1. In the abstract, there is too much background of work but little results. It should be modified.
2. Though the introduction is long, it is missing the function of “introduction”, which should be summarize the background of work as well as the literature review. It should be carefully re-written.
3. In Fig. 1, why is the chemical composition of tool steel in the range but not specific value? Are they measured on the parts used in the present work?
4. Though 560 C is designed to simulate the extrusion process, however, there is much higher force during the extrusion process, which could introduce much bigger change on the microstructure at this temperature. Authors should consider this point.
5. Authors should be carefully on the numbering the sections since they are all “ 1 “ for “introduction”, “Materials and methods”, “Results”, “Discussion” and “Conclusions”.
6. Please add the STD for Figs. 1, 4, and 10, etc.
7. Fig. 2 and Fig. 3 can be merged into 1.
8. How to identify the two layer with Fe2O3 and Fe3O4 with XRD? Did authors tried EBSD mapping?
9. In Fig. 8, the line scanning results in Fig. 8b is from which line in Fig. 8a?
10. The indicators for the phases in Fig. 9 is only partially shown.
11. In Fig. 10, though the hardness of V1 and V3 is higher than 1000, it is only 1023 and 1025. Considering the errors in measuring, it will be dangerous to meet the minimum requirement of 1000.
12. Please shorten the conclusion into few highlighted spots.
Author Response
Response to the Reviewer's 3 comments
All comments included in the review have been included in the revised article and marked in red.
The abstract has been corrected and redrafted. The introduction has been shortened, corrected and redrafted in line with the comments provided. The composition of the steel used for the tests has been given in article.
Table 1. The composition of the steel used for the tests
|
C |
Si |
Mn |
P |
S |
Cr |
Mo |
V |
|
0.39 |
0.99 |
0.31 |
0.018 |
0.003 |
5.07 |
1.22 |
0.93 |
Annealing at the temperature of 5900C simulated the extrusion process itself, which, depending on the test variant, lasted from 2 to 6 hours. The temperature of 5900C was adopted in the tests based on production data for the current year.
The article chapter numbering has been corrected.
X-ray examinations were carried out selectively at small angles to the surface (greasing), which made it possible to identify the chemical composition of the oxides layer. Investigations have been performed by using Bruker D8 ADVANCE X-ray diffractometer with filtered radiation Cu Kα (λ = 0.154056 nm) at room temperature. Test parameters were as follows: voltage – 40 kV, current -40 mA, angle 2Θ from 15◦ to 120◦.
Fig.8. Linear analysis of the chemical composition of the oxide layer, sample v7 (22 hours annealing), blue line – Fe, red line – O and green line – Si, the direction from left to right corresponds to the direction from the surface into the specimen, oxygen disappears after about 7 mm from surface, the green Si line in the area where oxygen is present indicates its presence in the oxidized layer
Fig.9 has been improved.
It was found that as a result of annealing, after various variants of the combination of temperatures of 4600C and 5900C, the hardness of most of the samples decreased below 1000 units. Only samples v1 and v3 after heat treatment showed a hardness above 1000 units, respectively sample v1 hardness 1023HV (with standard deviation 26,99) and sample v3 hardness 1025 HV (with standard deviation 11,22). Analyzing the obtained results, the hardness measurements were compared to the total annealing time of the samples, and it was found that the decrease in hardness correlates with the increase in the total annealing time at temperatures of 4600C and 5900C. The samples with hardness above 1000HV had the shortest annealing times. Respectively, the total annealing time of these samples was: for the sample v1 – 12 hours, and for the sample v3 – 6 hours. Taking into account the standard deviation, only the sample v3 with the lowest annealing time meets the hardness criterion.
The conclusions were shortened to the 3 most important statements.
The chemistry of the extruded aluminum alloy has been added to the Materials and Methods section.
The dies were used to extrude 6063 aluminum alloy sections with the composition given in Table 2. The ram speed of extrusion 6063 alloy process was 9mm/s.
Table 2. Chemical composition of the test melt from 6063 aluminum alloys
|
Alloy grade |
Si |
Fe |
Cu |
Mn |
Mg |
Zn |
Ti |
V |
|
6063 |
0,53 |
0,21 |
0,02 |
0,04 |
0,48 |
0,01 |
0,01 |
0,01 |
The tests showed that the samples v1 and v3 had the smallest diffusion layer width, and they were annealed at 4600C and 5900C, respectively, for 12 and 6 hours. Samples v1 and v3 were not kept at the temperature of 200C, which differs from the other samples. Additionally, they also had some of the shortest annealing times in 4600C and 5900C (Table 4.). Presumably, these factors limited the diffusion of nitrogen in the steel.
The two zones that make up the oxide layer are clearly visible. In literature it is often reported complex structure of oxide layers, which thickness increases with the annealing time [23,24,25]. The molecules of oxygen are absorbed from the atmosphere, reacted with the iron and nucleated of oxides formation of a thin layer which grows to a thicker layer with the increasing time of annealing.
The two-layer structure of the oxide on the surface of WNLV steel probably results from a multi-stage annealing at two temperatures of 4600C and 5900C. The kinetics of the growth of the oxide changes with increasing temperature, as well as its morphology. This causes changes in the chemical composition manifested by the diversity of structure growing oxide layers.
Drawings have been corrected in article text. An explanation has been added in Figure 8. The description of Rz in Fig. 12 has been changed. A speed framework is given in the Materials and Methods chapter. It was 9mm/s.
Deleted "research was carried out on the optimization of the annealing time" and inserted another description.
The introduction of additional elements in Fig. 1, 4, 10 will cause the loss of their legibility, therefore, please accept them as they are.

Round 2
Reviewer 3 Report
The paper quality has been greatly improved, especially the Introducion part. However, there are still few comments regarding to the modified manuscript:
(1) EBSD mapping is still suggested to show the two layers.
(2) It is still hard to be convinced by V3-6 to have a hardness higher than 1000 though it is explained by lower STD (11 ) at harndess of 1023.
(3) Please carefully check the text. For instance the followin text is not easy to understand:
"Only samples v1 and v3 after heat treatment showed a hard- 185 ness above 1000 units, respectively sample v1 hardness 1023HV (with standard deviation 186 26,99) and sample v3 hardness 1025 HV (with standard deviation 11,22).
